# 3D FEATURE PREDICTION FOR MASKED-AUTOENCODER-BASED POINT CLOUD PRETRAINING

**Siming Yan**[*†]**, Yuqi Yang**[‡]**, Yuxiao Guo**[‡]**, Hao Pan**[‡]
**Peng-Shuai Wang**[°]**, Xin Tong**[‡]**, Yang Liu**[‡]**, Qixing Huang**[†]
[†]The University of Texas at Austin, [‡]Microsoft Research Asia
[°]Peking University
{siming, huangqx}@cs.utexas.edu, {wangps}@hotmail.com
{t-yuqyan, Yuxiao.Guo, haopan, yangliu, xtong}@microsoft.com

## ABSTRACT

Masked autoencoders (MAE) have recently been introduced to 3D self-supervised pretraining for point clouds due to their great success in NLP and computer vision. Unlike MAEs used in the image domain, where the pretext task is to restore features at the masked pixels, such as colors, the existing 3D MAE works reconstruct the missing geometry only, i.e, the location of the masked points. In contrast to previous studies, we advocate that point location recovery is inessential and restoring intrinsic point features is much superior. To this end, we propose to ignore point position reconstruction and recover high-order features at masked points including surface normals and surface variations, through a novel attention-based decoder which is independent of the encoder design. We validate the effectiveness of our pretext task and decoder design using different encoder structures for 3D training and demonstrate the advantages of our pretrained networks on various point cloud analysis tasks. The code is available at https://github.com/SimingYan/MaskFeat3D.

## 1 INTRODUCTION

Self-supervised pretraining has recently gained much attention. It starts from a pretext task trained on large unlabeled data, where the learned representation is fine-tuned on downstream tasks. This approach has shown great success in 2D images (Chen et al., 2020; Grill et al., 2020; He et al., 2020; Bao et al., 2022; He et al., 2022; Zhou et al., 2022; Zhuang et al., 2021; 2019) and natural language processing (NLP) (Devlin et al., 2019; Brown et al., 2020) . Recently, people started looking into self-supervised pretraining on point cloud data due to its importance in 3D analysis and robotics applications.

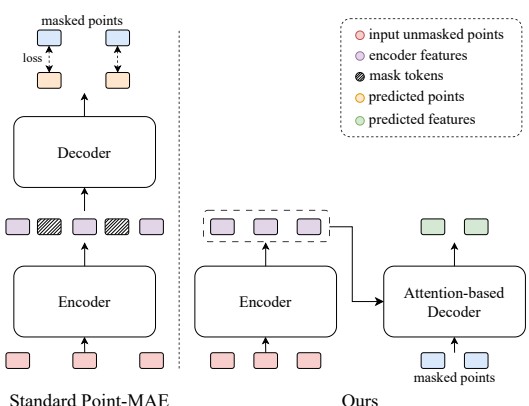

Figure 1: **Comparison of standard Point-MAE and our proposed method.** Unlike standard Point-MAE that uses masked points as the prediction target, our method use a novel attention-based decoder to leverage masked points as an additional input and infer the corresponding features.

An important self-supervised pretraining paradigm — *masked signal modeling* (MSM), including BERT (Bugliarello et al., 2021), BEiT (Bao et al., 2022), and masked autoencoders (MAE) (He et al., 2022), has recently been adopted to 3D domains. MSM has a simple setup: a randomly-masked input is fed to the encoder, and a decoder strives to recover the signal at the masked region. MSM is highly scalable and exhibits superior performance in many downstream vision and NLP tasks, outperforming their fully supervised equivalents. Additionally, it does not require extensive augmentation, which is essential and critical to another self-supervised

---

[*]Part of the work done when interning at Microsoft Research Asia.

pretraining paradigm — contrastive learning. In images, a mask refers to a randomly selected portion of the pixels, and the pixel colors or other pixel features in the masked region are to be reconstructed by the decoder.

For 3D point clouds, the PointBERT approach (Yu et al., 2022) masks point patches and recovers patch tokens that are pretrained by a point cloud Tokenizer. As reconstruction features are associated with patches of points, the learned features at the point level are less competitive. MAE-based pretraining schemes (Pang et al., 2022; Hess et al., 2022; Zhang et al., 2022; Liu et al., 2022) tackle this problem by point-wise pretext tasks. However, their decoders are designed to recover the positions of the masked points in Cartesian coordinates or occupancy formats (Fig. 1-left). **These designs make an intrinsic difference from 2D MSMs, where there is no need to recover masked pixel locations.** This key difference makes MSM pay more attention to capturing the irregular and possibly noisy point distribution and ignore the intrinsic surface features associated with points, which are essential for 3D point cloud analysis.

In the presented work, we propose to recover intrinsic point features, i.e., point normals, and surface variations (Pauly et al., 2002) at masked points, where point normals are first-order surface features and surface variations are related to local curvature properties. We clearly demonstrate that the recovery of high-order surface point features, not point locations, is the key to improving 3D MSM performance. Learning to reconstruct high-order geometric features forces the encoder to extract distinctive and representative features robustly that may not be captured by learning to reconstruct point positions alone. Our study justifies the importance of designing signal recovery for 3D MSMs. It aligns 3D MSM learning with MSM development in vision, where feature modeling plays a critical role (Wei et al., 2022).

To recover point signals, we design a practical attention-based decoder. This new decoder takes masked points as queries, and stacks several transformer blocks. In each block, self-attention is used to propagate context features over the masked points and cross-attention is applied to fabricate the point features with the encoder's output (As shown in Fig. 1-right and Fig. 2). This design is separable from the encoder design. Therefore, common 3D encoders, such as sparse CNNs, point-based networks, and transformer-based networks, can all be adopted to strengthen the pretraining capability. Another benefit of this decoder design is that the masked point positions are only accessible by the decoder, thus avoiding leakage of positional information in the early stage of the network, as suggested by (Pang et al., 2022).

We conducted extensive ablation studies to verify the efficacy of our masked feature design and decoder. Substantial improvements over previous approaches and the generalization ability of our pretraining approach are demonstrated on various downstream tasks, including 3D shape classification, 3D shape part segmentation, and 3D object detection. We hope that our study can stimulate future research on designing strong MAE-based 3D backbones.

We summarize the contributions of our paper as follows:

- We propose a novel masked autoencoding method for 3D self-supervised pretraining that predicts intrinsic point features at masked points instead of their positions.
- We introduce a unique attention-based decoder that can generate point features without relying on any particular encoder architecture.
- Our experiments demonstrate that restoring intrinsic point features is superior to point location recovery in terms of Point cloud MAE, and we achieve state-of-the-art performance on various downstream tasks.

## 2 RELATED WORK

**Self-supervised pretraining in 3D**  Self-supervised pretraining is an active research topic in machine learning (Liu et al., 2021). The early adoption of self-supervised pretraining for 3D is to use autoencoders (Yang et al., 2018; Yan et al., 2022) and generative adversarial networks (Wu et al., 2016) to learn shape-level features, mainly for shape classification and retrieval tasks. Other self-supervised pretext tasks, such as clustering and registration, are also developed for 3D pretraining. Later, due to the great ability to learn features at both the instance and pixel levels in a self-supervised manner, contrastive learning (Wu et al., 2018; Grill et al., 2020; He et al., 2020; Brown et al., 2020; Chen & He, 2021; Yan et al., 2023) was introduced into the 3D domains to extract distinctive instance

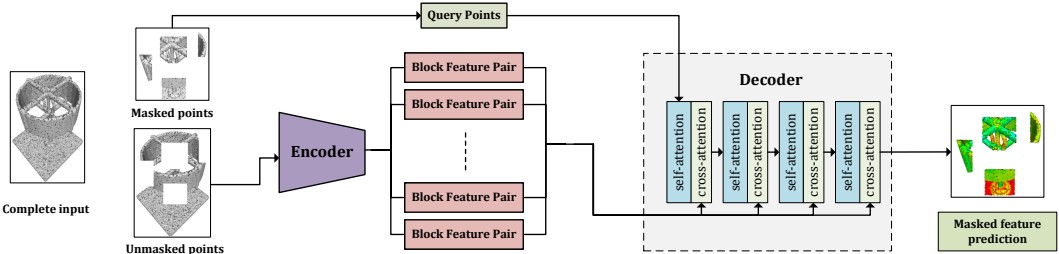

Figure 2: **The pretraining pipeline of our masked 3D feature prediction approach.** Given a complete input point cloud, we first separate it into masked points and unmasked points (We use cube mask here for better visualization). We take unmasked points as the encoder input and output the block feature pairs. Then the decoder takes the block feature pairs and query points(i.e., masked points) as the input, and predicts the per-query-point features.

and point-wise features for various downstream tasks (Wang et al., 2021b; Xie et al., 2020; Hou et al., 2021; Zhang et al., 2021). However, contrastive learning requires data augmentation heavily to form positive or negative pairs for effective feature learning.

**Masked signal modeling in 3D** Masked signal modeling using transformer-based architectures for self-supervised learning (SSL) has shown great simplicity and superior performance. PointBERT (Yu et al., 2022) and PointMAE (Pang et al., 2022) are two such works that inherit from this idea. PointBERT partitions a point cloud into patches and trains a transformer-based autoencoder to recover masked patches' tokens. In contrast, PointMAE directly reconstructs point patches without costly tokenizer training, using Chamfer distance as the reconstruction loss. Other works like (Zhang et al., 2022; Liu et al., 2022) and (Hess et al., 2022) explore different strategies for point cloud reconstruction or classification with masking. As discussed in Sec. 1, the pretext tasks of most previous works focus only on masked point locations.

**Signal recovery in masked autoencoders** Masked autoencoders for vision pretraining typically use raw color information in masked pixels as the target signal (He et al., 2022). However, Wei *et al.* (Wei et al., 2022) have found that using alternative image features, such as HOG descriptors, tokenizer features, and features from other unsupervised and supervised pretrained networks, can improve network performance and efficiency. In contrast, existing 3D MAE methods have limited use of point features and struggle with predicting the location of masked points. Our approach focuses on feature recovery rather than position prediction, selecting representative 3D local features such as point normals and surface variation (Pauly et al., 2002) as target features to demonstrate their efficacy. Our study allows for leveraging more advanced 3D features in 3D masked autoencoders, while further exploration of other types of 3D features (Laga et al., 2018) is left for future work.

## 3 MASKED 3D FEATURE PREDICTION

In this section, we present our masked 3D feature prediction approach for self-supervised point cloud pretraining. Our network design follows the masked autoencoder paradigm: a 3D encoder takes a point cloud whose points are randomly masked as input, and a decoder is responsible for reconstructing the predefined features at the masked points. The network architecture is depicted in Fig. 2. In the following sections, we first introduce the masking strategy and 3D masked feature modeling in Sec. 3.1 and 3.2, and then present our encoder and decoder design in Sec. 3.3 and 3.4. Here, the key ingredients of our approach are the design of prediction targets and the decoder, which govern the quality of the learned features.

### 3.1 3D MASKING

We follow the masking strategy proposed by PointBERT (Yu et al., 2022) to mask out some portions of an input point cloud and feed it to the encoder. Denote the input point cloud as $\mathcal{P} \in \mathbb{R}^{N \times 3}$, where $N$ is the number of points. We sample $K$ points using farthest point sampling (FPS). For each sample point, its $k$-nearest neighbor points form a point patch. For a given mask ratio $m_r, 0 < m_r < 1$, we randomly select $M$ patches and remove them from the input, where $M = \min(\lceil m_r \cdot K \rceil, K - 1)$.

In the following, the masked points and the remaining points are denoted by $\mathcal{P}_M$ and $\mathcal{P}_U$, respectively.

## 3.2 TARGET FEATURE DESIGN

As argued in Sec. 1, we advocate against using point locations as the reconstructed target. We choose to reconstruct normal and surface variation at each point, which reflect differential surface properties.

On the other hand, our decoder design (to be introduced in Sec. 3.4) takes query points as input and output predicted point-wise features. Therefore, the decoder implicitly carries positional information for learning meaningful features through the encoder.

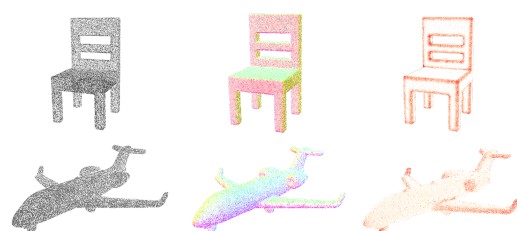

point cloud   point normal   surface variation

Given a point cloud, both point normal and surface variations are defined using local principal component analysis (PCA). We first define a covariance matrix $C_r$ over a local surface region around $\mathbf{p}$:

Figure 3: **Visualization of point features.**. The point normal is color-coded by the normal vector. The surface variation is color-coded where white indicates low value and red indicates high value.

$$C_r := \frac{\int_{\mathbf{x} \in \mathcal{S} \bigcap \mathbb{S}_r(\mathbf{p})} (\mathbf{p} - \mathbf{x})(\mathbf{p} - \mathbf{x})^T \, \mathrm{d}\mathbf{x}}{\int_{\mathbf{x} \in \mathcal{S} \bigcap \mathbb{S}_r(\mathbf{p})} \mathbf{1} \cdot \mathrm{d}\mathbf{x}}, \quad (1)$$

where $\mathcal{S} \bigcap \mathbb{S}_r(\mathbf{p})$ is the local surface region at $\mathbf{p}$, restricted by a sphere centered at $\mathbf{p}$ with radius $r$. We set $r = 0.1$ in our case. The ablation details are shown in the supplement.

The normal $\boldsymbol{n}(\mathbf{p})$ at $\mathbf{p}$ is estimated as the smallest eigenvector of $C_r$. The sign of each normal is computed by using the approach of (Hoppe et al., 1992).

Surface variation (Pauly et al., 2002) at $\mathbf{p}$ is denoted by $\sigma_r(\mathbf{p})$, in the following form:

$$\sigma_r(\mathbf{p}) = \frac{\lambda_1}{\lambda_1 + \lambda_2 + \lambda_3}, \quad (2)$$

where $\lambda_1 \leq \lambda_2 \leq \lambda_3$ are the eigenvalues of $C_r$. Surface variation is a geometric feature that measures the local derivation at point $\mathbf{p}$ in a neighborhood of size $r$ on a given surface $\mathcal{S}$ . Its original and modified versions have been used as a robust feature descriptor for a variety of shape analysis and processing tasks, such as saliency extraction (Pauly & Gross, 2001), curved feature extraction (Pauly et al., 2003), shape segmentation (Huang et al., 2006; Yan et al., 2021), and shape simplification (Pauly et al., 2002).

In the limit, i.e., when $r \to 0$, $\sigma_r(\mathbf{p})$ is related to the mean curvature (Clarenz et al., 2004). By varying the radii of $\mathbb{S}_r$, multiscale surface variation descriptors can be constructed. In our work, we chose only single-scale surface variation for simplicity.

Although both surface normal and surface variation are derived from local PCA, they are complementary to each other in the sense that surface normal carries first-order differential property while surface variation carries second-order differential property due to its relation to mean curvature. We visualize both features in Fig. 3 and show more examples in supplement. In Sec. 4.3, we show that reconstructing surface normal and surface variation leads to better learned features than reconstructing one of them.

**Loss function** Point normals and surface variations represent first- and second-order surface properties. Their value intervals are also bounded: *surface normal has unit length; surface variation is non-negative and not greater than $\frac{1}{3}$*. Their value-bounded properties are suitable for easy minimizing the deviation from the prediction to their ground truths, compared to using unbounded features such as curvatures. We denote the point normals and surface variations of $\mathcal{P}_M$ by $\mathcal{N}_M \in \mathbb{R}^{M \times 3}$ and $\mathcal{V}_M \in \mathbb{R}^M$, respectively. The loss function for pretraining the masked autoencoders is composed of two terms:

$$L_n = \|\mathcal{N}_M - \widehat{\mathcal{N}}_M\|_2^2; \quad (3)$$

$$L_v = \|\mathcal{V}_M - \widehat{\mathcal{V}}_M\|_1; \quad (4)$$

where $\widehat{\mathcal{N}}_M$ and $\widehat{\mathcal{V}}_M$ are the predicted versions of $\mathcal{N}_M$ and $\mathcal{V}_M$, respectively. The total loss function $L = \lambda_1 L_n + \lambda_2 L_v$, where $\lambda_1 = 1, \lambda_2 = 1$ in our case.

## 3.3 ENCODER DESIGN

Unlike most MAE-based approaches that are limited to ViT-based encoders, our approach is not restricted to any specific type of encoder. Common 3D encoders for point clouds are all supported, as long as the encoder outputs a set of learned features bind to spatial blocks, where spatial blocks could be point patches used for ViT-like transformer encoders (Yu et al., 2022; Pang et al., 2022; Liu et al., 2022; Zhang et al., 2022), set abstractions used by PointNet++-like encoders (Qi et al., 2017b; Qian et al., 2022), or coarse voxels used by sparse CNN-based encoders (Wang et al., 2017; Graham et al., 2018; Choy et al., 2019).

In the following, we briefly review these typical encoders and their adaption for our pretraining.

**ViT-based encoders**  These encoders first embed point patches via PointNet (Qi et al., 2017a), then send these patch tokens to a standard transformer that includes several multihead self-attention layers and feedforward layers. The transformer outputs the fabricated token features, corresponding to every input point patch. The token feature $\mathbf{f}_i$ and the patch center $\mathbf{c}_i$ form a block feature pair $B_i = \{\mathbf{f}_i, \mathbf{c}_i\}$, which is needed by our decoder. Here we can call $\mathbf{f}_i$ *block feature* and $\mathbf{c}_i$ *block centroid*.

**PointNet++-like encoders**  In these encoders, the network features are aggregated through a number of set abstraction levels. We take the learned features and the centroids at the coarsest set abstractions as block feature pairs.

**Sparse CNN-based encoders**  These encoders apply 3D convolution on sparse voxels from the finest level to the coarsest level. Multiple convolution layers and resblocks are commonly used. We interpolate the coarse voxel features at the centroids of the unmasked patches and use these interpolated features and the patch centroids to form our block feature pairs.

As suggested by (Pang et al., 2022), the early leaking of masked point information to the network could jeopardize feature learning. We adopt this suggestion: feed the unmasked points to the encoder only, and leave the masked points to the decoder.

## 3.4 DECODER DESIGN

**Decoder structure**  We design an attention-based decoder to restore the target features at masked regions. The decoder takes the block feature pairs $\mathcal{B} := \{B_i\}_{i=1}^b$ from the encoder and a query point set $\mathcal{Q}$, *i.e.*, the masked point set $\mathcal{P}_M$. It is composed of a stack of $l$ transformer blocks, where $l = 4$ in our case (See Fig. 2). Each block contains a self-attention layer and a cross-attention layer. The self-attention layer takes the query points and their positional embeddings as input and outputs the per-query point features, denoted by $\mathcal{S}^{in}$. Then $\mathcal{S}^{in}$ and the encoder block features $\mathcal{B}$ are passed into the cross-attention layer, where $\mathcal{S}^{in}$ serves as attention **query**, the block features serve as attention **key** and **value**, and the block centroids are the positional embedding of the block features. The output per-point features from the last block go through an MLP head to predict the target features at the query points.

**Efficacy of self-attention layers**  At first glance, it is sufficient to use cross-attention layers only for predicting per-point features. The recent masked discrimination work (Liu et al., 2022) obeys this intuition for its decoder design, no information exchanged between different query points. Instead, we introduce the self-attention layer to propagate information over query points and use multiple attention blocks to strengthen the mutual relationship progressively. We found that our design significantly improves feature learning, as verified by our ablation study (See Sec. 4.3).

**Supporting of various encoders**  In the above design, the decoder needs block feature pairs only from the encoder, thus having great potential to leverage various encoder structures, not limited to ViT-based transformer structures. This advantage is verified by our experiments (See Sec. 4).

**Feature reconstruction versus position reconstruction**  Note that our decoder and loss design do not explicitly model point positions, which are zero-order surface properties complementary to

surface normals and surface variations. Instead, the decoder predicts feature values at query points. Therefore, the zero-order positional information is already encoded implicitly. This explains why our approach is superior to baseline approaches that reconstruct point positions for feature learning (See Sec. 4.2).

**Query point selection** Due to the quadratic complexity of self-attention, the computational cost for a full query set could be much higher. In practice, we can randomly choose a point subset from $\mathcal{P}_M$ as the query set during training. By default, we use all masked points as queries.

# 4 EXPERIMENT ANALYSIS

We conducted a series of experiments and ablation studies to validate the efficacy and superiority of our masked 3D feature prediction approach, in short **MaskFeat3D**, for point cloud pretraining.

## 4.1 EXPERIMENT SETUP

**Pretraining dataset** We choose ShapeNet (Chang et al., 2015) dataset for our pretraining, following the practice of PointBERT (Yu et al., 2022) and previous 3D MAE-based approaches (Pang et al., 2022; Zhang et al., 2022; Liu et al., 2022). ShapeNet (Chang et al., 2015) contains 57 748 synthetic 3D shapes from 55 categories. We sample 50 000 points uniformly on each shape and select 128 nearest points from them for each point in the point cloud for constructing the local region to approximate surface variation. During pretraining, $N = 2048$ points are randomly sampled to create the point cloud.

**Network training** We integrated different encoders with our masked 3D feature prediction approach, including the ViT-based transformer used by (Pang et al., 2022), sparse-CNN-based encoder (Choy et al., 2019), and PointNeXt encoder (Qian et al., 2022) which is an advanced version of PointNet++. We implemented all pretraining models in PyTorch and used AdamW optimizer with $10^{-4}$ weight decay. We use PointBERT's masking strategy for ShapeNet pretraining. We set $K = 128$ in FPS, $k = 32$-nearest points to form the point patch, and the best masking ratio is $60\%$ empirically. The number of transformer blocks in the decoder is 4. The learning rates of the encoder and the decoder are set to $10^{-3}$ and $10^{-4}$, respectively. Standard data augmentation such as rotation, scaling, and translation are employed. All models were trained with 300 epochs on eight 16 GB Nvidia V100 GPUs. The total batch size is $64$.

**Downstream Tasks** We choose shape classification and shape part segmentation tasks to validate the efficacy and generalizability of our pretrained networks.

- **Shape classification**: The experiments were carried out on two different datasets: ModelNet40 (Wu et al., 2015) and ScanObjectNN (Uy et al., 2019). ModelNet40 is a widely used synthetic dataset that comprises 40 classes and contains 9832 training objects and 2468 test objects. In contrast, ScanObjectNN is a real-world scanned dataset that includes approximately 15 000 actual scanned objects from 15 classes. As the domain gap between ShapeNet and ScanObjectNN is larger than that between ShapeNet and ModelNet40, the evaluation on ScanObjectNN is a good measure of the generalizability of pretrained networks.
- **Shape part segmentation** ShapeNetPart Dataset (Yi et al., 2016) contains 16 880 models from 16 shape categories, and each model has 2∼6 parts. Following the standard evaluation protocol (Qi et al., 2017b), 2048 points are sampled on each shape. For evaluation, we report per-class mean IoU (cls. mIoU) and mean IoU averaged over all test instances (ins. mIoU).

The training-and-test split of the above tasks follows existing works. For these downstream tasks, we employ the task-specific decoders proposed by PointMAE (Pang et al., 2022) and reload the pretrained weights for the encoder. Training details are provided in the supplemental material.

## 4.2 EFFICACY OF 3D FEATURE PREDICTION

The advantages in learning discriminative features by our masked feature prediction approach are verified by its superior performance in downstream tasks.

| Method | ScanObjectNN | | | ShapeNetPart | | ShapeNetPart(1% labels) | |
|---|---|---|---|---|---|---|---|
| | OBJ-BG | OBJ-ONLY | PB-T50-RS | ins. mIoU | cls. mIoU | ins. mIoU | cls. mIoU |
| PointViT† Yu et al. (2022) | 79.9 | 80.6 | 77.2 | 85.1 | 83.4 | 77.6 | 72.2 |
| PointBERT Yu et al. (2022) | 87.4 | 88.1 | 83.1 | 85.6 | 84.1 | 79.2 | 73.9 |
| MaskDiscr Liu et al. (2022) | 89.7 | 89.3 | 84.3 | 86.0 | 84.4 | 78.8 | 72.3 |
| MaskSurfel Zhang et al. (2022) | 91.2 | 89.2 | 85.7 | 86.1 | 84.4 | - | - |
| PointMAE Pang et al. (2022) | 90.0 | 88.3 | 85.2 | 86.1 | - | 79.1 | 74.4 |
| MaskFeat3D (PointViT) | **91.7**(91.6) | **90.0**(89.6) | **87.7**(87.5) | **86.3**(86.3) | **84.9**(84.8) | **80.0**(79.9) | **75.1**(75.0) |

Table 1: **Performance comparison of MAE-based approaches on downstream tasks.** All the methods in the first section use the same transformer backbone architecture, PointViT. † represents the *from scratch* results and all other methods represent the *fine-tuning* results using pretrained weights. The average result of 3 runs is given in brackets.

| Method | ScanObjectNN | | | ShapeNetPart | |
|---|---|---|---|---|---|
| | OBJ-BG | OBJ-ONLY | PB-T50-RS | ins. mIoU | cls. mIoU |
| PointNet Qi et al. (2017a) | 73.3 | 79.2 | 68.0 | - | - |
| PointNet++ Qi et al. (2017b) | 82.3 | 84.3 | 77.9 | 85.1 | 81.9 |
| PointCNN Li et al. (2018) | 86.1 | 85.5 | 78.5 | 86.1 | 84.6 |
| DGCNN Wang et al. (2019) | 82.8 | 86.2 | 78.1 | 85.2 | 82.3 |
| MinkowskiNet Choy et al. (2019) | 84.1 | 86.1 | 80.1 | 85.3 | 83.2 |
| PointTransformer Zhao et al. (2021) | - | - | - | 86.6 | 83.7 |
| PointMLP Ma et al. (2022) | 88.7 | 88.2 | 85.4 | 86.1 | 84.6 |
| StratifiedTransformer Lai et al. (2022) | - | - | - | 86.6 | 85.1 |
| PointNeXt Qian et al. (2022) | 91.9 | 91.0 | 88.1 | 87.1 | 84.7 |
| MaskFeat3D (PointViT) | 91.7(91.6) | 90.0(89.6) | 87.7(87.5) | 86.3(86.3) | 84.9(84.8) |
| MaskFeat3D (MinkowskiNet) | 85.1(85.0) | 87.0(86.7) | 80.8(80.6) | 85.6(85.5) | 83.5(83.5) |
| MaskFeat3D (PointNeXt) | **92.7**(92.6) | **92.0**(91.9) | **88.6**(88.5) | **87.4**(87.4) | **85.5**(85.5) |

Table 2: **Comparison with supervised methods.** The average result of 3 runs is given in brackets.

**Comparison with MAE-based approaches** We compare our approach with other MAE-based approaches that use the same encoder structure. Tab. 1 reports that: (1) the performance of all MAE-based methods surpasses their supervised baseline – PointViT; (2) our strategy of reconstructing point features instead of point positions yields significant improvements in ScannObjectNN classification, improving overall accuracy on the most challenging split, PB-T50-RS, from $85.7\%$ (MaskSurfel) to $87.7\%$, and showing consistent improvements on other splits and ShapeNetPart segmentation.

We also compare the performance of our approach with PointBERT (Yu et al., 2022) ,PointMAE (Pang et al., 2022), and MaskDiscr (Liu et al., 2022) on ShapeNetPart segmentation with less labeled data. In this experiment, we randomly select $1\%$ labeled data from each category, and finetune the network with all selected data. The performance is reported in Tab. 1, which shows that using our pretrained network leads to much better performance than the baseline methods.

**Comparison with supervised approaches** Compared with state-of-the-art supervised methods, our approach again achieves superior performance than most existing works as seen from Tab. 2, including PointNet++ (Qi et al., 2017b) ,PointCNN (Li et al., 2018), DGCNN (Wang et al., 2019), MinkowskiNet (Choy et al., 2019), PointTransformer (Zhao et al., 2021) and PointMLP (Ma et al., 2022). It is only inferior to the approaches that use advanced encoder structures such as stratified transformer (Lai et al., 2022) and PointNeXt (Qian et al., 2022).

**Encoder replacement** To make a more fair comparison, we replaced the PointViT encoder with the PointNeXt's encoder, and retrained our pretraining network, denoted as MaskFeat3D (PointNeXt). From Tab. 2, we can see that our pretraining approach with this enhanced encoder can yield SOTA performance on all the downstream tasks, surpassing PointNeXt trained from scratch. We also used MinkowskiNet (Choy et al., 2019) as our pretraining encoder, the performance gain over MinkowskiNet trained from scratch is +0.7% overall accuracy improvement on ScanObjectNN classification, and +0.3% on ShapeNetPart segmentation. Please refer to the supplementary material for details.

**Few-shot Classification** To perform few-shot classification on ModelNet40, we adopt the "K-way N-shot" settings as described in prior work (Wang et al., 2021a; Yu et al., 2022; Pang et al., 2022). Specifically, we randomly choose K out of the 40 available classes and sample N+20 3D shapes per class, with N shapes used for training and 20 for testing. We evaluate the performance of MaskFeat3D under four few-shot settings: 5-way 10-shot, 5-way 20-shot, 10-way 10-shot, and 10-way 20-shot. To

| Method | 5-way | | 10-way | |
|---|---|---|---|---|
| | 10-shot | 20-shot | 10-shot | 20-shot |
| DGCNN[†] | $31.6 \pm 2.8$ | $40.8 \pm 4.6$ | $19.9 \pm 2.1$ | $16.9 \pm 1.5$ |
| OcCo | $90.6 \pm 2.8$ | $92.5 \pm 1.9$ | $82.9 \pm 1.3$ | $86.5 \pm 2.2$ |
| CrossPoint | $92.5 \pm 3.0$ | $94.9 \pm 2.1$ | $83.6 \pm 5.3$ | $87.9 \pm 4.2$ |
| Transformer[†] | $87.8 \pm 5.2$ | $93.3 \pm 4.3$ | $84.6 \pm 5.5$ | $89.4 \pm 6.3$ |
| OcCo | $94.0 \pm 3.6$ | $95.9 \pm 2.3$ | $89.4 \pm 5.1$ | $92.4 \pm 4.6$ |
| PointBERT | $94.6 \pm 3.1$ | $96.3 \pm 2.7$ | $91.0 \pm 5.4$ | $92.7 \pm 5.1$ |
| MaskDiscr | $95.0 \pm 3.7$ | $97.2 \pm 1.7$ | $91.4 \pm 4.0$ | $93.4 \pm 3.5$ |
| PointMAE | $96.3 \pm 2.5$ | $97.8 \pm 1.8$ | $92.6 \pm 4.1$ | $95.0 \pm 3.0$ |
| MaskFeat3D | $\mathbf{97.1 \pm 2.1}$ | $\mathbf{98.4 \pm 1.6}$ | $\mathbf{93.4 \pm 3.8}$ | $\mathbf{95.7 \pm 3.4}$ |

| Method | Target Feature | ScanNN |
|---|---|---|
| PointMAE | position only | 85.2 |
| | position + normal[*] | 85.7 |
| | position + surface variation[*] | 85.9 |
| | position + normal + variation[*] | 86.0 |
| MaskFeat3D | normal | 86.5 |
| | surface variation | 87.0 |
| | normal + variation | **87.7** |

Table 3: **Few-shot classification on ModelNet40.** We report the average accuracy (%) and standard deviation (%) of 10 independent experiments.

Table 4: **Ablation study on different features.** [*] uses position-index matching Zhang et al. (2022) for feature loss computation.

mitigate the effects of random sampling, we conduct 10 independent runs for each few-shot setting and report the mean accuracy and standard deviation. Additionally, more ModelNet40 results can be found in the supplementary material.

Overall, the improvements of our approach are consistent across different backbone encoders and datasets.

## 4.3 ABLATION STUDY

We proceed to present an ablation study to justify various design choices. For simplicity, we choose the shape classification task on ScanObjectNN, where the gaps under different configurations are salient and provide meaningful insights on the pros and cons of various design choices. Due to space constraints, additional ablation studies are available in the supplementary material.

**Decoder design** The primary question that arises is whether it is essential to disregard point position recovery. PointMAE's decoder follows a standard ViT-like architecture, utilizing a fully connected (FC) layer to directly predict the masked point coordinates. We implemented this decoder to predict our target features. However, since their decoder design does not encode masked point position, it cannot solely predict target features without predicting point position. To address this, we follow the approach proposed in (Zhang et al., 2022) and employ position-index matching for feature loss computation. As shown in Tab. 4, even though incorporating point features as the predicting target can enhance performance, the overall performance still significantly lags behind our design. This experiment highlights the significance of both point feature prediction and disregarding point position recovery.

**Target feature choice** In Tab. 4, the experiment shows that: (1) All combinations of point normal and surface variation can yield significant improvements over existing MAE approaches that recover point positions (*cf.* Tab. 1); (2) using both point normals and surface variations yields the best performance. As discussed in Sec. 3.2, this is due to the fact that they correspond to first- and second-order differential properties. They are relevant but complementary to each other. Therefore, reconstructing them together forces the encoder to learn more informative features than merely reconstructing one of them.

**Decoder depth** Tab. 5-a varies the number of transformer blocks (decoder depth). A sufficient deep decoder is necessary for feature learning. Increasing the number of blocks from 2 to 4 provides +1.5% improvement on ScanObjectNN classification task. The performance drops when increasing the depth further, due to the overfitting issue. Interestingly, we note that a 1-block decoder can strongly achieve 85.8% accuracy, which is still higher than the runner-up method (PointMAE).

**Data augmentation** Tab. 5-b studies three traditional data augmentation methods: rotation, scaling, and translation. Since the standard scaling could change the surface normal and variation, we scale the shape by using the same factor on 3 different axis. The experiments show that rotation and scaling play a more important role.

**Masking ratio.** Tab. 5-c varies the masking ratio of input point cloud, which is another important factor on our approach. When the masking ratio is too large, *e.g.*, 90%, the remaining part contains too limited information, which makes the task too hard to complete. When masking ratio is too

| (a) Decoder depth | | (b) Data augmentation | | | | (c) Mask ratio | | (d) Decoder attention | | (e) Query point ratio | |
|---|---|---|---|---|---|---|---|---|---|---|---|
| # blocks | ScanNN | rot | scale | trans | ScanNN | ratio | ScanNN | attention type | ScanNN | query/mask | ScanNN |
| 1 | 85.8 | √ | - | - | 87.0 | 40% | 86.8 | cross only | 85.7 | 25% | 85.7 |
| 2 | 86.2 | - | √ | - | 85.9 | 60% | **87.7** | cross+self | **87.7** | 50% | 86.2 |
| 4 | **87.7** | √ | √ | - | **87.7** | 90% | 86.5 | | | 75% | 86.6 |
| 8 | 87.5 | - | √ | √ | 85.1 | | | | | 100% | **87.7** |
| 12 | 87.1 | √ | √ | √ | 86.7 | | | | | | |

Table 5: **Ablation studies of our design choices**. Please refer to Sec. 4.3 for a detailed analysis.

small, *e.g.*, 40%, the task becomes too simple and impedes the feature learning. In our experiments, masking ratio=60% shows the best performance.

**Decoder block design** We tested whether the self-attention layer in our decoder is essential. By simply removing self-attention layers and using cross-attention layers only, we find that the performance has a large drop (-2.0), see Tab. 5-d.

**Number of query points** Finally, we varied the number of query points used by our decoder to see how it affects the network performance. Tab. 5-e shows that more query points lead to better performance. Here, "query/mask" is the ratio of selected query points with respect to the total number of masked points.

## 4.4 SCENE-LEVEL PRETRAINING EXTENSION

In principle, masked point cloud autoencoders could be scaled to noisy, large-scale point clouds. Additionally, we conducted an extension experiment on real-world scene-level data to evaluate our approach. Specifically, we pretrained our model on the ScanNet (Dai et al., 2017) dataset and evaluated its performance on 3D object detection task using the ScanNet and SUN RGB-D (Song et al., 2015) dataset. The training details can be found in the supplementary material. In this experiment, we observed that surface

| Method | Backbone | ScanNet | | SUN RGB-D | |
|---|---|---|---|---|---|
| | | $AP_{25}$ | $AP_{50}$ | $AP_{25}$ | $AP_{50}$ |
| STRL | VoteNet | 59.5 | 38.4 | 58.2 | 35.0 |
| RandomRooms | VoteNet | 61.3 | 36.2 | 59.2 | 35.4 |
| PointContrast | VoteNet | 59.2 | 38.0 | 57.5 | 34.8 |
| DepthContrast | VoteNet | 62.1 | 39.1 | 60.4 | 35.4 |
| Point-M2AE | Point-M2AE | 66.3 | 48.3 | - | - |
| MaskFeat3D | VoteNet | 63.3 | 41.0 | 61.0 | 36.5 |
| MaskFeat3D | Point-M2AE | 67.5 | 50.0 | - | - |
| MaskFeat3D | CAGroup3D | **75.6** | **62.3** | **67.2** | **51.0** |

Table 6: **3D object detection results.**

normal has a minor influence on the pretraining, while surface variation remains a robust feature. Moreover, we discovered that color signal could be an effective target feature. Hence, we pretrained our model with surface variation and color as the target features, and then fine-tuned the pretrained encoder on the downstream tasks. As shown in Tab. 6, given that previous studies lack a unified network backbone, we selected two of the most common works, VoteNet and Point-M2AE, along with the latest work, CAGroup3D, as the network backbones respectively. And our model exhibits consistent improvements in all the settings, which further proves the generalizability of our approach on noisy, large-scale point clouds. Although the concrete scene-level experiments are not the main focus of this paper, the results indicate that this is a promising direction.

## 5 CONCLUSION

Our study reveals that restoration of masked point location is not essential for 3D MAE training. By predicting geometric features such as surface normals and surface variations at the masked points via our cross-attention-based decoder, the performance of 3D MAEs can be improved significantly, as evaluated through extensive experiments and downstream tasks. Moreover, the performance gains remain consistent when using different encoder backbones. We hope that our study can inspire future research in the development of robust MAE-based 3D backbones.

**Acknowledgement.** Part of this work was done when Siming Yan was a research intern at Microsoft Research Asia. Additionally, we would like to acknowledge the gifts from Google, Adobe, Wormpex AI, and support from NSF IIS-2047677, HDR-1934932, CCF-2019844, and IARPA WRIVA program.

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
