# 3D Feature Prediction for Masked-AutoEncoder-Based Point Cloud Pretraining – Supplementary Material

Our supplementary materials provide implementation details of our model in Sec. 1, more experiment results in Sec. 2, more scene-level experiment results and training details in Sec. 3, more ablation studies in Sec. 4, qualitative results on target features in Sec. 5, and more visualization of shape surface variation in Sec. 6.

## 1 Implementation details

### 1.1 3D Masking

Given the input point cloud $\mathcal{P} \in \mathbb{R}^{N \times 3}$, by following the design of PointBERT (Yu et al., 2022), we first apply the Farthest Point Sampling algorithm for sampling $K$ center points. We group $k$ points for each center point via the K-Nearest-Neighborhood algorithm. Then we obtain $K$ point patches, and each point patch contains $k$ points. For a given mask ratio $m_r, 0 < m_r < 1$, we randomly select $M$ patches and remove them from the input, where $M = \min(\lceil m_r \cdot K \rceil, K - 1)$. The remaining number of patches $U = K - M$. In our experiment, we set $K = 128$, $k = 32$, and $m_r = 0.6$.

In the following, the masked points and the remaining points are denoted by $\mathcal{P}_M \in \mathbb{R}^{M \times k \times 3}$ and $\mathcal{P}_U \in \mathbb{R}^{U \times k \times 3}$, respectively.

### 1.2 Encoder structures

Our approach is not tied to any specific encoders. In our experiments, we apply three types of encoders: ViT-based encoder, PointNet++-like encoder, and Sparse CNN-based encoder. In this section, we show the implementation details of these encoders.

#### 1.2.1 ViT-based encoder

Following PointBERT (Yu et al., 2022), we use a standard transformer model as the encoder, which contains a group of transformer blocks. The input token sequence of the encoder consists of two parts: feature embedding and positional embedding:

**Feature embedding** We embed the unmasked point patches $\mathcal{P}_U \in \mathbb{R}^{U \times k \times 3}$ via a PointNet (Qi et al., 2017) network and output the feature embedding $T_F \in \mathbb{R}^{U \times C}$.

**Positional embedding** We first get the center points $\mathbf{c} \in \mathbb{R}^{U \times 3}$ of the unmasked point patches. Then we map $\mathbf{c}$ to the positional embedding $T_P \in \mathbb{R}^{U \times C}$ via a learnable MLP.

Then, we add $T_F$ and $T_P$ as the input to the transformer encoder.

The output of the transformer encoder $\mathbf{f} \in \mathbb{R}^{U \times C}$ shares the same shape as the input. Then we formulate the block feature pairs $B = \{\mathbf{f}, \mathbf{c}\}$ to further pass into our decoder. In our experiments, we set $C = 384$.

#### 1.2.2 PointNet++-like encoder

Because of its superior performance, we choose PointNeXt (Qian et al., 2022) as the PointNet++-like encoder backbone. PointNeXt proposed an advanced version of PointNet++ by improving the data augmentation and optimizing the training strategies.

In detail, given the unmasked point patches $\mathcal{P}_U \in \mathbb{R}^{U \times k \times c}$, we first reshape the point patches into one point cloud $\mathcal{P}_{U'} \in \mathbb{R}^{U' \times c}$, where $U' = U \times k$. Taken $\mathcal{P}_{U'}$ as the input, PointNeXt hierarchically abstracts features of the input by using Set Abstraction(SA) and Feature Propagation(FP) layer from PointNet++. The SA layer downsamples the input point cloud, and the FP layer aggregates the features. In our experiments, we use four blocks during the pretraining and set the downsampling rate to 2. Then we get the output of the PointNeXt encoder $\mathbf{f} \in \mathbb{R}^{\frac{U'}{16} \times C}$, where $C = 512$ in our case. We denote $\mathbf{c} \in \mathbb{R}^{\frac{U'}{16} \times 3}$ as the corresponding center points of $\mathbf{f}$. Similarly, we formulate the block feature pairs $B = \{\mathbf{f}, \mathbf{c}\}$ to further pass into our decoder.

| cases | channel(c) | ScanObjectNN | ShapeNetPart |
|---|---|---|---|
| Scratch | 3 | 87.1 | 85.6 |
| MaskFeat3D | 3 | 88.0 | 86.8 |
| Scratch | 4 | 88.1 | 86.0 |
| MaskFeat3D | 4 | **88.6** | 87.0 |
| Scratch | 7 | - | 87.1 |
| MaskFeat3D | 7 | - | **87.4** |

Table 1: **Comparison of PointNeXt backbone under different input channel numbers.** 'Scratch' means training from scratch. 'MaskFeat3D' means fine-tuning on the pretrained weight of our model. We report overall accuracy for ScanObjectNN PB-T50-RS split, and ins. mIoU for ShapeNetPart task. Channel $c = 3$ means only using position. $c = 4$ uses position and height information. $c = 7$ uses position, height, and point normal. Since ScanObjectNN dataset does not contain point normal labels, we leave $c = 7$ entries blank.

In the general case, the input channel number $c = 3$ represents the position of points, which is the same as other encoder backbones. However, in the downstream tasks, PointNeXt concatenates height dimension($c = 4$) for the ScanObjectNN classification task and concatenates both normal and height dimension($c = 7$) for the ShapeNetPart segmentation task(Please refer to (Qian et al., 2022) for more details). Therefore, to better transfer the pretraining knowledge, we decide to keep the same dimension as the downstream tasks during the pretraining: We set $c = 4$ for the pretraining model transferring to the ScanObjectNN classification task, and set $c = 7$ for the pretraining model transferring to the ShapeNetPart segmentation task.

However, frequently switching the channel number of input will harm the flexibility of the pretraining model, e.g., the pre-trained model with $c = 4$ cannot directly transfer to the tasks required input channel $c = 7$ because of the mismatched input shape. Therefore, we experimented that when the input channels are all the same, our pre-trained weight could still benefit all the downstream tasks. The results are shown in Tab. 1. Fine-tuning on our pretrained weights shows consistent improvement under different input channel numbers.

### 1.2.3 SPARSE CNN-BASED ENCODER

We choose MinkowskiNet (Choy et al., 2019) as the Sparse CNN-based encoder backbone.

In the pretraining stage, the unmasked point patches are reshaped to $\mathcal{P}_{U'} \in \mathbb{R}^{U' \times c}$ like Sec. 1.2.2. Then we voxelized the point cloud, using voxel size $v = 0.03125$. The encoder hierarchically extracts features from the sparse voxel input. Each level contains two sparse CNN-based red blocks and a convolution layer whose stride and kernel size are 2 to downsample the features. The encoder finally outputs the voxel features whose stride is 8. We interpolate the output voxel features with patch centers using trilinear interpolation to obtain the block feature $\mathbf{f}$. Similarly, we formulate the block feature pairs $B = \{\mathbf{f}, \mathbf{c}\}$ to further pass into our decoder.

We apply a global MAX POOLING layer after the encoder for the classification tasks to get the shape feature. We interpolate the voxel features whose strides are 2,4,8 using the input point cloud for the segmentation task. Then we concatenate these features to get the per-point features for the segmentation prediction.

### 1.3 TRAINING DETAILS

In the pre-training stage, we keep the same training strategy for all the different encoders. We use AdamW optimizer with $10^{-4}$ weight decay. All models were trained with 300 epochs.

| Method | ModelNet40 |
|---|---|
| PointViT[†] Yu et al. (2022) | 91.4 |
| PointBERT Yu et al. (2022) | 93.2 |
| MaskDiscr Liu et al. (2022) | 93.8 |
| MaskSurfel Zhang et al. (2022) | 93.6 |
| PointMAE Pang et al. (2022) | 93.8 |
| MaskFeat3D (PointViT) | **93.9** |

Table 2: **Shape classification fine-tuned on ModelNet40.** All the methods use the same transformer backbone architecture. [†] represents the *from scratch* results and all other methods represent the *fine-tuning* results using pretrained weights.

| Method | Acc. (%) |
|---|---|
| 3D-GAN Wu et al. (2016) | 83.3 |
| Latent-GAN Achlioptas et al. (2018) | 85.7 |
| SO-Net Li et al. (2018) | 87.3 |
| MAP-VAE Han et al. (2019) | 88.4 |
| Jigsaw Sauder & Sievers (2019) | 84.1 |
| FoldingNet Yang et al. (2018) | 88.4 |
| DGCNN + OcCo Wang et al. (2021) | 89.7 |
| DGCNN + STRL Huang et al. (2021) | 90.9 |
| PointViT + OcCo[*] Wang et al. (2021) | 89.6 |
| PointBERT Yu et al. (2022)[*] | 87.4 |
| PointMAE Pang et al. (2022)[*] | 88.5 |
| MaskFeat3D (PointViT) [*] | **91.1** |

Table 3: **Linear evaluation for shape classification on ModelNet40.** This task is sensitive to the encoder backbone. Different * methods use the same Transformer encoder backbone.

In the fine-tuning stage for different downstream tasks, we share the same training strategy as the training from scratch.

## 2 MORE EXPERIMENT RESULTS

### 2.1 SHAPE CLASSIFICATION ON MODELNET40

In this section, we evaluate our method by supervised fine-tuning on ModelNet40 dataset. As shown in Tab. 2, under the same Transformer backbone, PointViT, our method shows the best performance.

### 2.2 LINEAR SVM ON MODELNET40

We also evaluate the performance of different pretrained networks by fixing the pretrained weights and using linear SVMs on ModelNet40 classification (see Tab. 3). Compared with other MAE-based approaches that use the same ViT-based encoder structure, our approach achieves the best classification accuracy.

### 2.3 FEATURE VISUALIZATION

We also assess the discriminativity of learned features by visualizing point features as follows. For a given point on a point cloud, we linearly interpolate block features at this point, using the inverse of its distance to the block centroids as the interpolation weights. We then project all the interpolated point features into the 3D space, via T-SNE embedding (van der Maaten & Hinton, 2008). The 3D space is treated as the RGB color space for assigning colors to points. Here, the block features are from the pretrained encoder, not finetuned on downstream tasks. The color difference between different points characterizes their feature differences, and distinguishable features are preferred by many downstream tasks. The first column of Fig. 1 shows the point features of two chair shapes learned by our approach using the ViT-based encoder.

We also visualize the point features learned by PointBERT (Yu et al., 2022) and other MAE-based pretraining approaches: PointMAE (Pang et al., 2022), MaskDiscr (Liu et al., 2022), MaskSurfel (Zhang et al., 2022), where all these methods use a similar ViT-based encoder. Fig. 1 shows that our learned features are more discriminative than other methods. For instance, the legs of the chair in the first row from our method — MaskFeat3D, are more distinguishable than the results of PointBert and MaskSurfel, because of their clear color difference; the two arms and different legs in the second row from MaskFeat3D are also more discriminative than those from other methods.

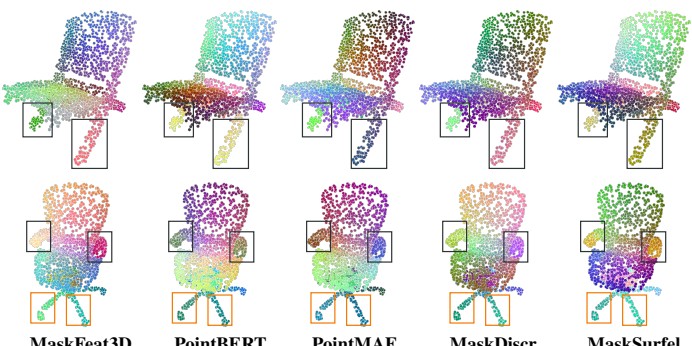

Figure 1: **Feature visualization of different pretrained approaches** including our MaskFeat3D, PointBERT Yu et al. (2022), PointMAE Pang et al. (2022), MaskDiscr Liu et al. (2022), MaskSurfel Zhang et al. (2022). Here, note that the absolute colors from different methods are not comparable as their feature spaces are not the same, instead the point color difference from the same method is a good visual measurement for assessing feature discriminativity. First row: By comparing the colors of the two front legs, we can see our pretrained network produces more discriminative features than PointBERT and MaskSurfel. Second row: It is clear the features on the different chair legs from our methods, as well as the features of two chair arms, are more discriminative.

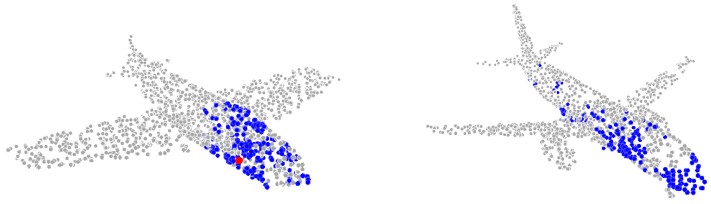

Figure 2: Illustration of zero-shot correspondence learning.

## 2.4 ZERO-SHOT CORRESPONDENCE LEARNING

We also study how our model's learned feature aids zero-shot correspondence learning. As shown in Fig. 2, we randomly selected a query point (red) on the object to the left and showed the nearest 300 points (blue) in the feature space on that object and another one from the same category. We observed that the nearest points shared similar semantic information across both objects, thereby demonstrating the efficacy of our model's feature learning.

## 3 MORE SCENE-LEVEL EXPERIMENT RESULTS

### 3.1 PRETRAINING DETAILS

We performed extension experiments using real-world scene-level data to assess our model's capability on noisy, large-scale point clouds. Specifically, we first pre-trained our model on the ScanNet (Dai et al., 2017) dataset, which comprises over 190k RGB-D data from approximately 1,513 indoor scenes. To generate colored point clouds, we mapped the depth images to world coordinates with the RGB signal. We utilized these point clouds as our pre-training data, and pretrained the model for 300 epochs.

### 3.2 S3DIS DATASET RESULTS

We also evaluated the performance of our pre-trained model on 3D object detection and indoor semantic segmentation using the S3DIS (Armeni et al., 2016) dataset. This dataset comprises 3D scans of 272 rooms from six buildings, with 3D instance and semantic annotations. For the object detection task, we evaluated our method on furniture categories, with AABBs derived from 3D

| Method | Sem Seg | | Detection | |
|---|---|---|---|---|
| | mIoU | OA | mAP$_{0.25}$ | mAP$_{0.5}$ |
| PointNeXt[†] Qian et al. (2022) | 70.8 | 90.7 | - | - |
| MaskFeat3D | **71.7** | **91.7** | - | - |
| FCAF3D[†] Rukhovich et al. (2022) | - | - | 66.7 | 45.9 |
| MaskFeat3D | - | - | **71.6** | **49.2** |

Table 4: **Area 5 Semantic segmentation and detection results on S3DIS.** [†] represents the *from scratch* results and MaskFeat3D in the same section represents the *fine-tuning* results using pretrained weights under same backbone.

| (a) **Neighborhood size** | | (b) **Number $k$ in FPS** | | (c) **Mask region** | |
|---|---|---|---|---|---|
| r | ScanObjectNN | # $k$ | ScanObjectNN | region | ScanObjectNN |
| 0.05 | 86.6 | 8 | 87.4 | masked only | **87.7** |
| 0.07 | 87.0 | 16 | 87.4 | all points | 86.8 |
| 0.1 | **87.7** | 32 | **87.7** | | |
| 0.2 | 86.8 | 64 | 87.6 | | |
| | | 128 | 87.6 | | |

Table 5: **More ablation studies of our design choices**. Please refer to Sec. 4 for more details.

semantics. We used the official split, with 68 rooms from Area 5 reserved for validation, while the remaining 204 rooms comprised the training subset.

To train our model, we utilized the most recent publicly available approaches. Specifically, we employed FCAF3D (Rukhovich et al., 2022) as the training architecture for the object detection task, and PointNeXt (Qian et al., 2022) for the semantic segmentation task. In the fine-tuning stage, we adopted the same training strategy as that used for training from scratch.

As shown in Tab. 4, our model exhibits consistent improvements in both tasks, which further proves the generalizability of our approach on noisy, large-scale point clouds.

# 4 MORE ABLATION STUDY RESULTS

## 4.1 ABLATION STUDY ON SHAPE SURFACE VARIATION

Surface variation is a geometric feature that measures the local derivation at point **p** in a neighborhood of size $r$ on a given surface $\mathcal{S}$. In this section, we study the influence of neighborhood of size $r$.

In our experiments, we use ShapeNet as the pre-training dataset. Each shape in ShapeNet is normalized to $[-1, 1]$. For constructing the local region to approximate surface variation, we select $k$ nearest points for each point and compute the mean distance as the neighborhood size $r$. As shown in Tab. 5a, we set the neighborhood size to be 0.05, 0.07, 0.1, 0.2, which correspond to 32, 64, 128, and 512 nearest points, respectively. We observe that $r = 0.1, k = 128$ yields the best performance. It is also worth noting that all different settings demonstrate better performance than the prior state-of-the-art method (MaskSurfel (Zhang et al., 2022), 85.7%).

## 4.2 ABLATION STUDY ON NUMBER K IN FPS

In our masking strategy, we denote the input point cloud as $\mathcal{P} \in \mathbb{R}^{N \times 3}$, where $N$ is the number of points. And we sample $K = 128$ points using farthest point sampling (FPS). For each sample point, its $k$-nearest neighbor points form a point patch. The $k$ controls the size of point patch, which is important to the pretraining. As shown in Tab. 5b, our experiment shows that $k = 32$ gives the best result.

## 4.3 ABLATION STUDY ON UNMASKED PATCHES

In our experiment, we only predicted the target features of query points at the masked region. It is also worthy to note if we select the query points from all point patches, including both masked and unmasked regions. As shown in Tab. 5c, only selecting the query points from masked region shows better performance.

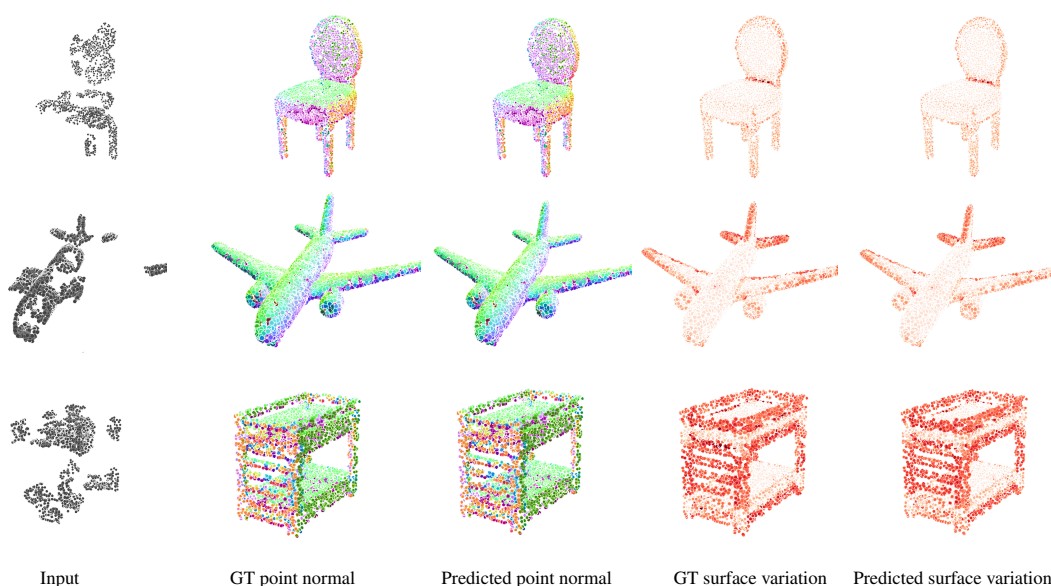

| Input | GT point normal | Predicted point normal | GT surface variation | Predicted surface variation |

Figure 3: **Qualitative results on target features.** The masking ratio of input point cloud is 60%.

## 5 QUALITATIVE RESULTS ON TARGET FEATURES

In this section, we present qualitative results for our target features, namely point normals and surface variation. As depicted in Fig. 3, our model demonstrates accurate predictions for both of these features.

## 6 MORE VISUALIZATION OF SHAPE SURFACE VARIATION

In this section, we show more visualization results of shape surface variation in the ShapeNet dataset. We set $r = 0.1$ for all shapes. As shown in the Fig. 4, we can observe that shape surface variation can capture various geometric feature signals of different shapes.

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

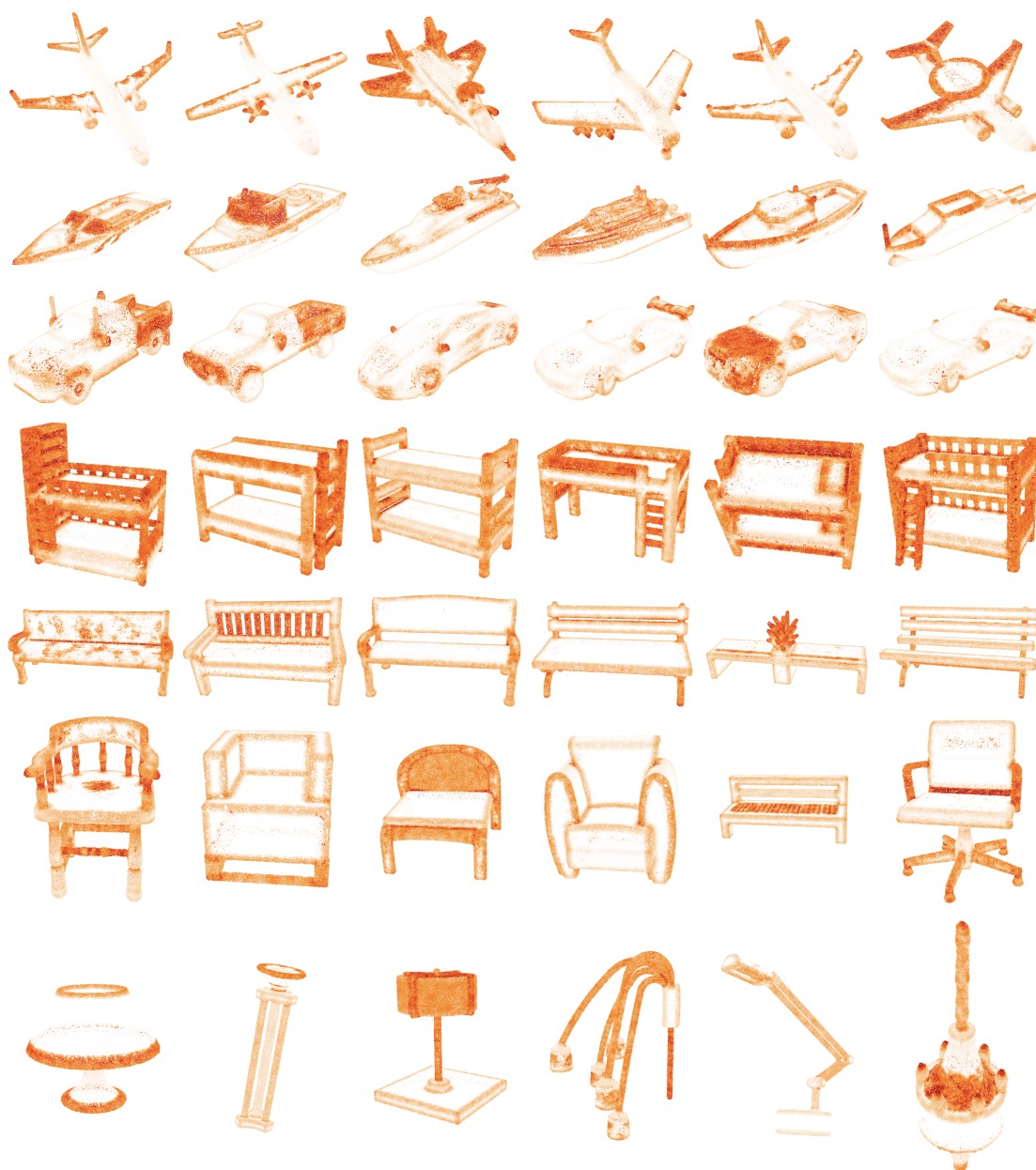

Figure 4: **Visualization results of shape surface variation in the ShapeNet dataset.** Red means high values and white means low values.

Siyuan Huang, Yichen Xie, Song-Chun Zhu, and Yixin Zhu. Spatio-temporal self-supervised representation learning for 3d point clouds. In *Proceedings of the IEEE/CVF International Conference on Computer Vision*, pp. 6535–6545, 2021.

Jiaxin Li, Ben M Chen, and Gim Hee Lee. So-net: Self-organizing network for point cloud analysis. In *Proceedings of the IEEE conference on computer vision and pattern recognition*, pp. 9397–9406, 2018.

Haotian Liu, Mu Cai, and Yong Jae Lee. Masked discrimination for self-supervised learning on point clouds. In *ECCV*, 2022.

Yatian Pang, Wenxiao Wang, Francis E. H. Tay, W. Liu, Yonghong Tian, and Liuliang Yuan. Masked autoencoders for point cloud self-supervised learning. In *ECCV*, 2022.

Charles R Qi, Hao Su, Kaichun Mo, and Leonidas J Guibas. PointNet: Deep learning on point sets for 3D classification and segmentation. In *CVPR*, pp. 652–660, 2017.

Guocheng Qian, Yuchen Li, Houwen Peng, Jinjie Mai, Hasan Abed Al Kader Hammoud, Mohamed Elhoseiny, and Bernard Ghanem. PointNeXt: Revisiting PointNet++ with improved training and scaling strategies. In *NeurIPS*, 2022.

Danila Rukhovich, Anna Vorontsova, and Anton Konushin. Fcaf3d: fully convolutional anchor-free 3d object detection. In *Computer Vision–ECCV 2022: 17th European Conference, Tel Aviv, Israel, October 23–27, 2022, Proceedings, Part X*, pp. 477–493. Springer, 2022.

Jonathan Sauder and Bjarne Sievers. Self-supervised deep learning on point clouds by reconstructing space. *Advances in Neural Information Processing Systems*, 32, 2019.

Laurens van der Maaten and Geoffrey Hinton. Visualizing data using t-sne. *Journal of Machine Learning Research*, 9(86):2579–2605, 2008.

Hanchen Wang, Qi Liu, Xiangyu Yue, Joan Lasenby, and Matt J Kusner. Unsupervised point cloud pre-training via occlusion completion. In *ICCV*, pp. 9782–9792, 2021.

Jiajun Wu, Chengkai Zhang, Tianfan Xue, Bill Freeman, and Josh Tenenbaum. Learning a probabilistic latent space of object shapes via 3d generative-adversarial modeling. *Advances in neural information processing systems*, 29, 2016.

Yaoqing Yang, Chen Feng, Yiru Shen, and Dong Tian. Foldingnet: Point cloud auto-encoder via deep grid deformation. In *Proceedings of the IEEE conference on computer vision and pattern recognition*, pp. 206–215, 2018.

Xumin Yu, Lulu Tang, Yongming Rao, Tiejun Huang, Jie Zhou, and Jiwen Lu. Point-BERT: Pre-training 3D point cloud transformers with masked point modeling. In *CVPR*, pp. 19313–19322, June 2022.

Yabin Zhang, Jiehong Lin, Chenhang He, Yongwei Chen, Kui Jia, and Lei Zhang. Masked surfel prediction for self-supervised point cloud learning. arXiv:2207.03111, 2022.