# OpenReview forum: "3D Feature Prediction for Masked-AutoEncoder-Based Point Cloud Pretraining"
_ICLR.cc/2024/Conference — ICLR 2024 poster_

### Official Review · Reviewer_UgQs · 2023-10-30

**Soundness:** 3 good
**Presentation:** 3 good
**Contribution:** 2 fair
**Rating:** 6
**Confidence:** 4

**Summary:**

In this paper, the authors apply the self-supervised pretraining paradigm of masked signal modeling to point cloud pretraining. They propose a novel approach called MaskFeat3D, which focuses on recovering high-order features of masked points rather than their locations. Additionally, they propose an encoder-agnostic attention-based decoder. The effectiveness of the proposed method is evaluated through experiments conducted on the ScanObjectNN dataset for shape classification and the ShapeNetPart dataset for shape part segmentation.

**Strengths:**

- The authors present evidence that the recovery of high-order point features yields more effective results compared to the recovery of point positions for 3D masked signal modeling.
- A novel encoder-agnostic attention-based decoder is proposed by the authors to accurately regress the high-order features of masked points.
- The paper is well-written and provides clear explanations, making it easy to follow.

**Weaknesses:**

- It appears that this is not the first work in 3D masked signal modeling that focuses on recovering high-order features of masked points. For example, MaskSurfel (Zhang et al.) specifically aims to recover surface normals of points. This similarity with previous work may diminish the novelty of the paper.
- The results on the ScanObjectNN dataset indicate that Point-MA2E outperforms MaskFeat3D significantly; however, this comparison is not included in the paper.
- To assess the effectiveness of the decoder, it is recommended to include results obtained by combining the block features and masked queries, and feeding them into self-attention blocks of the same depth, similar to the MAE approach, with points used as the positional embeddings.


[1] Zhang et al., Point-MA2E: Masked and Affine Transformed AutoEncoder for Self-supervised Point Cloud Learning.

**Questions:**

See Weaknesses.

---

> ### Author Response · Authors · 2023-11-13
> **Official Comment by Authors (Part 1)**
>
> We extend our heartfelt thanks to the reviewer for their insightful feedback. In addressing the concerns raised, we are pleased to provide the following clarifications and additional information.
>
> **Q: Not the first work on recovering high-order features of masked points.**
>
> We thank the reviewer for pointing this out. We argue that there are key differences when comparing our method with MaskSurfel:
>
> 1. From the perspective of insights, our approach, as the reviewer wYee notes, emphasizes the importance of ignoring the recovery of irregular and potentially noisy point positions, and focusing on intrinsic surface features for effective 3D MSM pre-training. In contrast, MaskSurfel predicts not only surface normals but also **point positions**. Furthermore, the surface normal prediction in MaskSurfel depends on the position prediction.
> 2. From the perspective of model design, we propose a novel encoder-agnostic attention-based decoder specifically aimed at ignoring point position and concentrating solely on intrinsic features. Conversely, MaskSurfel made an extenion from Point-MAE. The decoder design of MaskSurfel is the same as Point-MAE.
> 3. From the perspective of experiment result, our method shows better results than MaskSurfel. As detailed in Table 4 of the main paper (referenced below), MaskSurfel's result is categorized as “position + normal" under the PointMAE section. As a fair comparison, our method taking only surface normal as the target feature ("normal" under MaskFeat3D section) also shows a significant improvement compared with MaskSurfel. This experiment further proves the validity of our insights and the effectiveness of our decoder design.
>
> ----
>
>
>
> | Method     | Target Feature                 |      | ScanObjectNN PB-T50-RS |
> | ---------- | ------------------------------ | ---- | ---------------------- |
> | PointMAE   | position only                  |      | 85.2                   |
> |            | **position + normal**∗         |      | **85.7**               |
> |            | position + surface variation∗  |      | 85.9                   |
> |            | position + normal + variation∗ |      | 86.0                   |
> | MaskFeat3D | **normal**                     |      | **86.5**               |
> |            | surface variation              |      | 87.0                   |
> |            | normal + surface variation     |      | 87.7                   |
>
> **Table 4: Ablation study on different features.** ∗ uses position-index matching Zhang et al.[1] for feature loss computation.
>
> ----
>
>
>
> **Q: Point-MA2E shows better results.**
>
> We are grateful to the reviewer for mentioning Point-MA2E, which introduces an intriguing concept of improving the masking strategy with affine transformation. However, it is important to clarify the idea of Point-MA2E is orthogonal to our method. Point-MA2E is dedicated to refining the masking strategy, while our work focuses on studying the pre-training pretext task. We speculate that Point-MA2E's mask strategy might improve our method, and we will provide a discussion on this in our revised manuscript.
>
> Additionally, **the Point-MA2E paper has not yet been published, and its supplementary material remains unavailable**. This restricts our ability to fully grasp their evaluation procedures and conduct further systematical comparisons.
>
>
>
> [1] Zhang, Yabin, et al. "Masked surfel prediction for self-supervised point cloud learning." *arXiv preprint arXiv:2207.03111* (2022).

---

> > ### Author Response · Authors · 2023-11-13
> > **Official Comment by Authors (Part 2)**
> >
> > **Q: Assess the effectiveness of the decoder.**
> >
> > We are thankful to the reviewer for their insightful suggestion. In response, we conducted this comparison experiment. We removed the cross-attention module, directing both masked queries and block features into the self-attention blocks, which is similar to MAE approach. The results of this comparison are presented in the table below, showing that our method achieves consistent improvements over the suggested alternative design.
> >
> > | Method         | Target Feature             | ScanObjectNN PB-T50-RS |
> > | -------------- | -------------------------- | ---------------------- |
> > | Self-attention | normal                     | 85.9                   |
> > |                | surface variation          | 86.2                   |
> > |                | normal + surface variation | 86.9                   |
> > | MaskFeat3D     | normal                     | 86.5                   |
> > |                | surface variation          | 87.0                   |
> > |                | normal + surface variation | 87.7                   |
> >
> > We hypothesize that the removal of the cross-attention module might have increased the complexity of the learning process. This complexity arises as the decoder needs to simultaneously process its own previous outputs and the encoder’s outputs. Such a scenario could potentially lead to information confusion. The cross-attention mechanism serves a crucial role in distinctly separating information originating from the decoder itself and that from the encoder. Blending these two types of information could hinder the model's ability to differentiate between relationships in the encoder features and query points.

---

> ### Comment · Area_Chair_vg52 · 2023-11-22
>
> Reviewer UgQs,
>
> Has the rebuttal addressed your concerns? Please reply to the authors, and post your decision ASAP.
>
> AC

---

> > ### Comment · Reviewer_UgQs · 2023-11-22
> > **Post-rebuttal assessment**
> >
> > I appreciate the authors' responses which sufficiently address my concerns. I also observed that they conducted more experiments on the semantic segmentation task in their response to Reviewer HyKn, which demonstrates the versatility of their methods. Consequently, I would revise my rating to a positive one.

---

> > > ### Author Response · Authors · 2023-11-22
> > > **Thank you for your response!**
> > >
> > > Thank you for your feedback and for improving the score! We are glad that our responses have addressed your concerns! Thank you again for the time and effort you have dedicated to reviewing our work!

---

### Official Review · Reviewer_cRKn · 2023-10-30

**Soundness:** 3 good
**Presentation:** 3 good
**Contribution:** 3 good
**Rating:** 8
**Confidence:** 4

**Summary:**

The paper proposes a point cloud pre-training method to improve the downstream tasks’ performances. More specifically, instead of predicting point positions by a masked autoencoder, the authors propose to recover high-order features at masked points including surface normals and surface variations through a novel attention-based decoder. To verify the effectiveness of the method, various point cloud analysis tasks have been tested, and promising results have been achieved.

**Strengths:**

1. The idea is interesting, and the results are promising.
2. Extensive experiments are conducted with SOTA performances.
3. The paper is clearly written and well-organized.

**Weaknesses:**

It seems that the ablation study shown in Table 4 failed to support the idea that it is essential to disregard point position recovery, since at the same time to predict point positions using PointMAE, the decoder architecture is also changed when using MaskFeat3D architecture. To make a fairer comparison, the same decoder architecture from MaskFeat3D should be used to predict point position as well.

**Questions:**

Since the authors claim that it is essential to disregard point position recovery. Hence, how to understand that predicting point positions actually enhances the performances when using PointMAE in Table 4?

---

> ### Author Response · Authors · 2023-11-13
>
> We deeply appreciate your insightful and positive comments about our contributions to the community.
>
> **Q: Predict point position under the same archtecture of MaskFeat3D.**
>
> We thank the reviewer for pointing this out. Since our method has already taken the point position as an additional input for the decoder, it is trivial to use our decoder architecture to predict the point position. The model would simply be learning to replicate the input it was already given. Following the reviewer's suggestion, we conducted this experiment, and the results are presented below. We found that only predicting point positions does not lead to any improvement in downstream tasks. Furthermore, we experimented additional experiments with  'position + normal', 'position + surface variation', and 'position + normal + surface variation'.  We observe that adding position prediction to our method does not yield any improvement. These results prove that incorporating position prediction to our method is unnecessary.
>
> ----
>
> | Target Feature                        | ScanObjectNN **PB-T50-RS** |
> | ------------------------------------- | -------------------------- |
> | Train from scratch                    | 77.2                       |
> | position only                         | 77.5                       |
> | position + normal                     | 86.4                       |
> | normal                                | 86.5                       |
> | position + surface_variation          | 87.1                       |
> | surface variation                     | 87.0                       |
> | position + normal + surface variation | 87.6                       |
> | normal + surface variation            | 87.7                       |
>
> **Table: Ablation study on the MaskFeat3D architecture.** All experiments were conducted using the same MaskFeat3D decoder architecture. The term 'Train from scratch' denotes no pretraining procedure.
>
> ----
>
>
>
> **Q: How to understand that predicting point positions actually enhances the performances?**
>
> We are grateful for the reviewer's inquiry, although it does bring some confusion on our end. The primary objective of Table 4 is twofold: to demonstrate that integrating intrinsic 3D features can indeed enhance the efficacy of the Point-MAE framework, and to highlight that, even with these enhancements, their performance remains markedly inferior to that achieved by our decoder design.
>
> It is crucial to clarify that the results in Table 4 do not suggest that predicting point positions enhances performance. Furthermore, as we mentioned in the previous question, our experimental findings indicate that focusing on the prediction of point positions under our decoder architecture does not lead to any improvements in performance.

---

> > ### Comment · Reviewer_cRKn · 2023-11-21
> > **Thank you for the rebuttal.**
> >
> > The rebuttal has resolved my concerns, and I keep my positive rating.

---

> > > ### Author Response · Authors · 2023-11-21
> > > **Thank you for your positive rating!**
> > >
> > > Thank you for your positive rating! We are glad that our responses have addressed your concerns!
> > >
> > > Thank you again for the time and effort you have dedicated to reviewing our work!

---

### Official Review · Reviewer_wYee · 2023-10-31

**Soundness:** 2 fair
**Presentation:** 2 fair
**Contribution:** 2 fair
**Rating:** 6
**Confidence:** 4

**Summary:**

This paper proposes a self-supervised learning method from point cloud. Typically, this paper addresses the importance of using surface normal and surface variance instead of using point location as proposed by the previous studies. The idea is straightforward and easy-to-understand. The experiments demonstrate that the efficacy of the proposed method. Moreover, the ablation study consistently proves the addressed issue by the authors.

**Strengths:**

The authors address the importance of the geometric measurements for the usage of pre-training the network. Typically, using surface normal as surface variation are meaningful in point cloud based understanding. Typically, the authors provide the various experiments such as backbone architectures, loss designs, and downstream task evaluations. I really enjoyed reading this paper.

**Weaknesses:**

There are some minor things that need to be discussed

W-1. Analysis of 2D/3D masked autoencoders.

In the manuscript, the authors commented that __"These designs make an intrinsic difference from 2D MSMs, where there is no need to recover masked pixel locations."__

It is true. I understand the analysis by the authors. When we think of the vanilla MAE, it also takes a masked image as an input and predicts the color information, not its pixel location. However, when we think of the nature of the point cloud, it is sparse, irregular, and unordered. Even, I would say _raw point cloud_ naturally does not involve color information. Accordingly, it is not feasible to extend the concept of the MAE for the 2D image into the MAE for the 3D points. In my opinion, the authors should have written such a clear understanding of MAE for 3D points.

W-2. Details in computing surface normal and surface variance on scene-level experiments.
While the various object-level datasets, such as shapenet, are synthetically created, the real-world points are captured by the sensors. Due to such difference, raw point cloud from the real world naturally involves lots of noise, which could be an issue when computing surface normal using PCA. So I wonder how the authors solve this issue when conducting experiments on Sec. 4-4 in the manuscripts.

W-3. Insightful analysis
I truly agree that the proposed experiments demonstrate that the surface normal and surface variance are important measurements for self-supervising learning using 3D points. Technically, I also agree with such an observation. However, I wonder why such an approach brings performance improvement. Is there any geometric analysis? Based on the manuscript, this approach can be viewed as a naive extension of the Point-MAE that additionally uses other geometric measurements.

I want to know the author's own analysis of such problem setup and insights.

**Questions:**

Alongside with the addressed weakness, I have one minor question.

Q-1. __Is there any reason that authors did not conduct experiments on the S3DIS dataset using 3D semantic segmentation?
If there are some reasonable and meaningful results, I can convince the efficacy of this work. Otherwise, this work could be understood as naive extension.__

**Details Of Ethics Concerns:**

There is no ethic issues.

---

> ### Author Response · Authors · 2023-11-13
> **Official Comment by Authors (Part 1)**
>
> We extend our heartfelt thanks to the reviewer for the valuable feedback. In addressing the concerns raised, we are pleased to provide the following clarifications and additional information.
>
> **W-1: Analysis of 2D/3D masked autoencoders.**
>
> We are grateful for the insightful suggestion and deeply appreciate the reviewer's accurate understanding of our insight. We agree that, unlike grid-organized 2D image data, point cloud data is irregular and possibly noisy, making direct recovery of point positions less effective in capturing the intrinsic features of 3D shapes. We will clarify our statement more explicitly in our revision.
>
> **W-2: Surface normal is noisy in real world scenes.**
>
> We thank the reviewer for this valuable feedback. During our experiments, we also observed that surface normals in scene data are noisy and often lack consistent orientation due to incomplete point clouds. However, we found that surface variation remains robust in these conditions. Therefore, as mentioned in Section 4.4, we took the rgb color signal and surface variation as the target features, rather than the surface normal.
>
> **W-3-1: Naive extension of Point-MAE**.
>
> We appreciate the reviewer for raising this important concern. We would like to argue that our method is not merely a simple extension of Point-MAE.
>
> Firstly, as elaborated in Section 1 and illustrated in Figure 1, in addition to the insight mentioned by the reviewer in W-1 (which we believe is also an important contribution), we introduce a general pre-training framework with a novel attention-based decoder, thereby enhancing the pre-training model. Our model is not restricted to any specific encoder design and is easily adaptable to various task settings, including those in challenging indoor environments. Also, by disregarding point positions and focusing on intrinsic 3D shape features, our model learns robust features suitable for diverse downstream tasks.
>
> Secondly, as mentioned in Table 4 of the main paper (referenced below), a naive extension of Point-MAE with additional features does not yield the same level of significant improvement as our method. This finding further proves the validity of our insights and the effectiveness of our decoder design.
>
> ----
>
>
>
> | Method     | Target Feature                 |      | ScanObjectNN PB-T50-RS |
> | ---------- | ------------------------------ | ---- | ---------------------- |
> | PointMAE   | position only                  |      | 85.2                   |
> |            | position + normal∗             |      | 85.7                   |
> |            | position + surface variation∗  |      | 85.9                   |
> |            | position + normal + variation∗ |      | 86.0                   |
> | MaskFeat3D | normal                         |      | 86.5                   |
> |            | surface variation              |      | 87.0                   |
> |            | normal + surface variation     |      | 87.7                   |
>
> **Table 4: Ablation study on different features.** ∗ uses position-index matching Zhang et al.[1] for feature loss computation.
>
> ----
>
> **W-3-2: More analysis.**
>
> In the supplementary material, we provide further experimental analysis. In Section 2.3, we visualized the point features of our model and other MAE-based approaches. Our approach shows more discriminative features than other methods. This observation proves that learning to reconstruct high-order geometric features robustly enables the encoder to extract more distinctive and representative features.
>
> In Section 2.4 of the supplementary material, we investigated how the learned features of our model assist in zero-shot correspondence learning, showing that points nearest in the feature space exhibit similar semantic information, even across different objects within the same class. We hope these analyses further highlight the robustness and effectiveness of our pre-training strategy.
>
> [1] Zhang, Yabin, et al. "Masked surfel prediction for self-supervised point cloud learning." *arXiv preprint arXiv:2207.03111* (2022).

---

> > ### Author Response · Authors · 2023-11-13
> > **Official Comment by Authors (Part 2)**
> >
> > **Q-1: Experiments on semantic segmentation tasks.**
> >
> > We thank the reviewer for pointing this out. We did conduct semantic segmentation task on S3DIS Area 5 using the PointNeXt [1] encoder backbone. These results are detailed in Table 4 of the supplementary material and summarized below. We observed a significant improvement of +0.9/+1 in mean Intersection over Union (mIoU) and Overall Accuracy (OA) compared to the baseline training from scratch. This outcome underlines the potential efficacy of our pre-training approach.
> >
> > ----
> >
> > | Method     | S3DIS Area 5 | Semantic Seg |      |      | S3DIS Area 5 | Detection   |
> > | ---------- | ------------ | ------------ | ---- | ---- | ------------ | ----------- |
> > |            | mIoU         | OA           |      |      | mAP$_{0.25}$ | mAP$_{0.5}$ |
> > | PointNeXt† | 70.8         | 90.7         |      |      | -            | -           |
> > | MaskFeat3D | **71.7**     | **91.7**     |      |      | -            | -           |
> > | FCAF3D†    | -            | -            |      |      | 66.7         | 45.9        |
> > | MaskFeat3D | -            | -            |      |      | **71.6**     | **49.2**    |
> >
> > **Supp Table 4: Area 5 Semantic segmentation and detection results on S3DIS.** † represents the *from scratch* results and MaskFeat3D in the same section represents the *fine-tuning* results using pretrained weights under same backbone.
> >
> > ----
> >
> >
> >
> > To further validate the effectiveness of our method, we have conducted additional experiments. As we claimed that our method is encoder-agnostic, we experimented with another commonly used encoder backbone, Sparse-UNet [2]. During pre-training, we maintained identical settings, only replacing the encoder backbone with Sparse-UNet.  The results of this experiment are as follows:
> >
> > | Method             | Backbone        | S3DIS Area 5 mIoU | S3DIS 6-Fold mIoU | ScanNet mIoU |
> > | ------------------ | --------------- | ----------------- | ----------------- | ------------ |
> > | Train from scratch | Sparse-UNet     | 68.2              | 73.6              | 72.2         |
> > | PointContrast      | Sparse-UNet     | 70.9              | -                 | 74.1         |
> > | DepthContrast      | Sparse-UNet     | 70.6              | -                 | 71.2         |
> > | **MaskFeat3D**     | **Sparse-UNet** | **72.3**          | **76.4**          | **74.7**     |
> >
> > These results, spanning S3DIS Area 5, S3DIS 6-Fold, and ScanNet datasets, consistently show improvements over training from scratch and outperform other pre-training strategies under the same encoder backbone. This evidence supports the effectiveness of our method for more challenging downstream tasks.
> >
> >
> >
> > [1] Qian, Guocheng, et al. "Pointnext: Revisiting pointnet++ with improved training and scaling strategies." In NeurIPS 2022.
> >
> > [2] Choy, Christopher, JunYoung Gwak, and Silvio Savarese. "4D spatio-temporal convnets: Minkowski convolutional neural networks." In CVPR 2019.

---

> ### Comment · Reviewer_wYee · 2023-11-21
> **Thank you for the rebuttal.**
>
> __Overall, I am quite positive about this paper.__ Though it could be understood as a naive extension using surface normal and normal variance. For me, it is good enough.
>
> The experiments are conducted in object-scale datasets as well as room-scale datasets, which makes it reasonable for me to understand the benefit of using surface normal as self-supervision. I hope that the authors will put these results in the final manuscript as well as the supplementary material.
>
> Also, __I recommend the authors to modify the title.__ This is not a pioneering work that applies MAE to point cloud understanding. However, the title can mislead the readers. I think that the main contribution comes from using surface normal as a self-supervision signal. __While the authors insist on the novelty of the decoder network design, it looks trivial.__ It is not a big deal. Accordingly, the titles should involve the terminologies, such as surface normal, intrinsics points, or point normal, etc.
>
> __Change rate: 5 --> 6: marginally above the acceptance threshold__

---

> > ### Author Response · Authors · 2023-11-21
> > **Thank you for your response!**
> >
> > Thank you for your insightful feedback and for improving the score! We deeply appreciate your recommendation and will give them thorough consideration. We will deliberate on a more fitting title that accurately reflects the core contribution of our work.

---

### Official Review · Reviewer_HyKn · 2023-10-31

**Soundness:** 2 fair
**Presentation:** 3 good
**Contribution:** 2 fair
**Rating:** 6
**Confidence:** 4

**Summary:**

The paper proposes a pre-training task for 3D encoders, so, later, can lead to improved performance when fine-tuned on a downstream task. The pre-training objective is the prediction of point-surface properties such as normal or surface variation from masked regions of the input point cloud.

**Strengths:**

The paper proposes an alternative to point coordinates prediction on a mask auto-encoder setup. Sampling point coordinates can be difficult for decoder architectures as the ones used by previous works. However, by fixing the point coordinate in the decoder these problems disappear and the task becomes to predict shape properties around the queried point.

**Weaknesses:**

I like the main idea of the paper, it is well presented and presents a significant improvement over previous works for most of the task. However, my main concern is not only related to this work in particular but to this line of works where they focus on tasks related to single objects. I have been playing around with these datasets for many years already, and I can say that datasets such as classification on ModelNet40, and segmentation on ShapeNet are relatively "easy", there is a lot of noise in the annotations, and I believe the improvements presented by current methods is simply overfitting to this specific data set. In other subfields of computer vision, a pre-training paper that is only evaluated on MNIST or CIFAR10 would not be accepted, but for some reason, they do for point clouds. So, I don't find these works convincing since the reported results and architectures usually do not translate to more challenging tasks such as semantic segmentation or instance segmentation on real 3D scans. That being said, this work presents results on the task of object detection of ScanNet and SUN-RGBD, which I believe is the right direction. However, I think more results reported on other tasks such as semantic or instance segmentation should be necessary to determine the quality of the pre-training strategy. Therefore, I will rate this paper marginally below the acceptance but I will be happy to see additional results during the rebuttal phase.

**Questions:**

I would encourage the authors to include more challenging tasks such as semantic and instance segmentation of 3D scans.

---

> ### Author Response · Authors · 2023-11-13
>
> We express our sincere gratitude to the reviewer for the insightful feedback. In response to the concerns raised, we offer the following clarifications and additional information.
>
> **Q: This field only focuses on tasks related to single objects.**
>
> Firstly, we deeply appreciate the reviewer highlighting this phenomenon. It is indeed true that many studies in the point cloud field have primarily concentrated on shape-level tasks, which may not always be the optimal choice to evaluate pre-training strategies. This is also one of the reasons why we extended our method to more challenging tasks such as 3D object detection. However, as the reviewer mentioned, most works in the point cloud field have not pursued this direction.
>
> Actually, we have conducted the semantic segmentation task on S3DIS Area 5 using the PointNeXt [1] encoder backbone in our submission. These results are detailed in Table 4 of supplementary material and summarized below. We observed a significant improvement of +0.9/+1 in mean Intersection over Union (mIoU) and Overall Accuracy (OA) compared to the baseline training from scratch. This outcome underlines the potential efficacy of our pre-training approach.
>
> ----
>
> | Method     | S3DIS Area 5 | Semantic Seg |      |      | S3DIS Area 5 | Detection   |
> | ---------- | ------------ | ------------ | ---- | ---- | ------------ | ----------- |
> |            | mIoU         | OA           |      |      | mAP$_{0.25}$ | mAP$_{0.5}$ |
> | PointNeXt† | 70.8         | 90.7         |      |      | -            | -           |
> | MaskFeat3D | **71.7**     | **91.7**     |      |      | -            | -           |
> | FCAF3D†    | -            | -            |      |      | 66.7         | 45.9        |
> | MaskFeat3D | -            | -            |      |      | **71.6**     | **49.2**    |
>
> **Supp Table 4: Area 5 Semantic segmentation and detection results on S3DIS.** † represents the *from scratch* results and MaskFeat3D in the same section represents the *fine-tuning* results using pretrained weights under same backbone.
>
> ----
>
>
>
> To further validate the effectiveness of our method, we conducted additional experiments for the rebuttal. As we claimed that our method is encoder-agnostic, we experimented with another commonly used encoder backbone, Sparse-UNet [2]. During pre-training, we maintained identical settings, only replacing the encoder backbone with Sparse-UNet.  The results of this experiment are as follows:
>
> | Method             | Backbone        | S3DIS Area 5 mIoU | S3DIS 6-Fold mIoU | ScanNet mIoU |
> | ------------------ | --------------- | ----------------- | ----------------- | ------------ |
> | Train from scratch | Sparse-UNet     | 68.2              | 73.6              | 72.2         |
> | PointContrast      | Sparse-UNet     | 70.9              | -                 | 74.1         |
> | DepthContrast      | Sparse-UNet     | 70.6              | -                 | 71.2         |
> | **MaskFeat3D**     | **Sparse-UNet** | **72.3**          | **76.4**          | **74.7**     |
>
> These results, spanning S3DIS Area 5, S3DIS 6-Fold, and ScanNet datasets, consistently show improvements over training from scratch and outperform other pre-training strategies under the same encoder backbone. This evidence supports the effectiveness of our method for more challenging downstream tasks in indoor environments.
>
> [1] Qian, Guocheng, et al. "PointNext: Revisiting pointnet++ with improved training and scaling strategies." In NeurIPS 2022.
>
> [2] Choy, Christopher, JunYoung Gwak, and Silvio Savarese. "4D spatio-temporal convnets: Minkowski convolutional neural networks." In CVPR 2019.

---

> > ### Comment · Reviewer_HyKn · 2023-11-22
> > **Post-rebuttal assessment**
> >
> > The authors have included additional experiments on more complex tasks according to my suggestions. Moreover, they pointed out additional results that were also in the supplementary material of the original submission. Therefore, all my concerns have been addressed. I increased my score from 5 to 6.
> > One small remark that I think could improve the paper is to include the supplementary material as an appendix in the main paper. This will allow readers to analyze these additional experiments directly in the paper.

---

> > > ### Author Response · Authors · 2023-11-22
> > > **Thank you for your response!**
> > >
> > > Thank you for your response and for improving the score! In accordance with your suggestion, we will integrate our supplementary material with the main paper to enhance the overall reading experience!

---

> ### Comment · Area_Chair_vg52 · 2023-11-22
>
> Reviewer HyKn,
>
> Has the rebuttal addressed your concerns? Please reply to the authors, and post your decision ASAP.
>
> AC

---

### Author Response · Authors · 2023-11-21

We appreciate all the reviewers for their hard work! Please find our responses to your individual questions below. We look forward to discussing any issues further should you have any follow-up concerns!

---

### Meta-Review · Area_Chair_vg52 · 2023-12-09

**Metareview:**

The papers describes a novel approach in 3D self-supervised pretraining using masked autoencoders (MAEs), focusing on recovering intrinsic point features like surface normals and variations instead of point location, through an attention-based decoder that enhances performance on various point cloud analysis tasks. All reviewers agree to accept the paper. The reviewers generally find the paper interesting and the results support the paper's claim. The AC agree with the reviewers on accepting the paper.

**Justification For Why Not Higher Score:**

The experiments can still be improved.

**Justification For Why Not Lower Score:**

All reviewers agree on acceptance.

---

### Decision · Program_Chairs · 2024-01-16

Accept (poster)